# Glioma *biopsies* Classification Using Raman Spectroscopy and Machine Learning Models on Fresh Tissue Samples

**DOI:** 10.3390/cancers13051073

**Published:** 2021-03-03

**Authors:** Marco Riva, Tommaso Sciortino, Riccardo Secoli, Ester D’Amico, Sara Moccia, Bethania Fernandes, Marco Conti Nibali, Lorenzo Gay, Marco Rossi, Elena De Momi, Lorenzo Bello

**Affiliations:** 1Department of Medical Biotechnology and Translational Medicine, Università Degli Studi di Milano, 20122 Milan, Italy; 2Unit of Oncological Neurosurgery, Humanitas Clinical and Research Center—IRCCS, 20089 Rozzano, Italy; tommaso.sciortino@unimi.it (T.S.); marco.conti@unimi.it (M.C.N.); lorenzo.gay@unimi.it (L.G.); marco.rossi2@unimi.it (M.R.); lorenzo.bello@unimi.it (L.B.); 3Department of Oncology and Hemato-Oncology, Università degli Studi di Milano, 20122 Milano, Italy; 4The Hamlyn Centre for Robotic Surgery, Institute of Global Health Innovation, Imperial College London, London SW7 2AZ, UK; r.secoli@imperial.ac.uk; 5Department of Electronics, Information and Bioengineering, Politecnico di Milano, Piazza Leonardo da Vinci 32, 20133 Milano, Italy; ester.damico@mail.polimi.it (E.D.); elena.demomi@polimi.it (E.D.M.); 6The BioRobotics Institute and Department of Excellence in Robotics and AI, Scuola Superiore Sant’Anna, Piazza Martiri della Liberta’ 33, 56127 Pisa, Italy; s.moccia@staff.univpm.it; 7Unit of Pathology, Humanitas Clinical and Research Center—IRCCS, via Manzoni 56, 20089 Rozzano, Italy; bethania.fernandes@humanitas.it

**Keywords:** Raman spectroscopy, neuro-oncology, classification, glioma, machine learning

## Abstract

**Simple Summary:**

Raman Spectroscopy (RS) is an optical technique able to determine biochemical differences within a biological tissue, yet there are few reports of its application in fresh tissue. This potential could be relevant to detect glioma infiltration and to guide surgery to achieve a total resection. We deployed RS with machine learning algorithms in a surgical scenario to probe the diagnostic accuracy *ex-vivo* in discriminating between normal-appearing and fresh neoplastic tissue in patients undergoing glioma resection. Analyzing 3450 spectra from 63 samples, we identified 19 novel RS shifts. These 19 novel shifts of known biological relevance were included in the analytical workflow, leading to 83% and 82% accuracy and precision, respectively, in discriminating between normal-appearing and neoplastic tissue. This study further supported the translational development of real-time tissue analysis with RS in oncological brain surgery and yielded novel findings in the biochemical features of the brain tissue affected by glioma.

**Abstract:**

Identifying tumor cells infiltrating normal-appearing brain tissue is critical to achieve a total glioma resection. Raman spectroscopy (RS) is an optical technique with potential for real-time glioma detection. Most RS reports are based on formalin-fixed or frozen samples, with only a few studies deployed on fresh untreated tissue. We aimed to probe RS on untreated brain biopsies exploring novel Raman bands useful in distinguishing glioma and normal brain tissue. Sixty-three fresh tissue biopsies were analyzed within few minutes after resection. A total of 3450 spectra were collected, with 1377 labelled as Healthy and 2073 as Tumor. Machine learning methods were used to classify spectra compared to the histo-pathological standard. The algorithms extracted information from 60 different Raman peaks identified as the most representative among 135 peaks screened. We were able to distinguish between tumor and healthy brain tissue with accuracy and precision of 83% and 82%, respectively. We identified 19 new Raman shifts with known biological significance. Raman spectroscopy was effective and accurate in discriminating glioma tissue from healthy brain ex-vivo in fresh samples. This study added new spectroscopic data that can contribute to further develop Raman Spectroscopy as an intraoperative tool for in-vivo glioma detection.

## 1. Introduction

Gliomas have an infiltrative growth pattern. A key challenge is to discriminate tumor and healthy tissue during surgery, and to identify neoplastic cells infiltrating beyond evident pathological borders [1]. Identifying the residual tumor burden is crucial as the total resection represents a favorable prognostic factor [2,3]. Unlike extracranial oncological surgery procedures, the intra-axial masses cannot be removed by applying large safety margins based only on visible landmarks, due to the proximity to eloquent structures. Damaging functional tissue could cause neurological deficits with a negative impact on the subject [4]. The definition itself of tumor border is also demanding as glioma cells are known to infiltrate normal-appearing white matter and be undetectable with current diagnostic methods in-vivo [5]. Imaging and fluorescence-guidance are currently used to delineate the boundaries of the resection. However, these methods have limited sensitivity, specificity and spatial resolution in vivo [6].

Raman spectroscopy (RS), and other optical techniques such as Fourier Transform Infrared Spectroscopy (FTIR) [7,8], emerged as a tool to study the biochemical differences between tumor and healthy tissue. The rapid and non-destructive features of RS made it an engaging method to detect tumor cells intraoperatively. A quick, unbiased brain tumor detection technique during glioma resection is of paramount relevance to achieve additional quantitative diagnostic intra-operative information.

Previous studies investigated the use of the RS to discriminate between human and animal brain tissue [8,9,10,11,12,13] in vitro and in-vivo [8] and tumor and healthy tissue on sections obtained from preprocessed tissue blocks [8]. However, tissue processing (de-waxing, washing, and clearing with solvents) can affect tumors’ biochemical phenotype by altering different cellular lipid constituents [14]. Changes following a chemical treatment can alter the spectral signatures and reduce the spectral differences between normal and malignant tissue [15,16]. Only a few studies are available with data from fresh tissue [5,17,18] or in-vivo [11]. Most studies are performed on frozen or Formalin-Fixed Paraffin Preserved (FFPP) samples with known artefacts [19]. The use of fresh tissue obtained during surgery and analyzed right after excision with engineering tools [18] for spectra analysis was thus advocated for improving Raman measurements and ease clinical translation.

We aimed to evaluate RS’s ability to distinguish unaffected tissue from glioma ex-vivo on fresh tissue samples from biopsies of supratentorial glioma and to explore whether additional Raman peaks could be detected onto untreated brain tissue.

## 2. Materials and Methods

### 2.1. Experimental Design

This study deployed single-point Raman spectroscopy on 63 biopsies from adult subjects undergoing surgery for WHO grade II-IV gliomas. The number of spectra and histological groups are reported in (Table 1). The ethics committee approved the study (Study n. 595) performed according to the 1964 Declaration of Helsinki and later amendments. Safe supra-marginal resection was pursued as previously described [20] with imaging and neurophysiological guidance [6,21].

### 2.2. Sample Preparation and Analysis

The bulk of the tumor tissue was sent for pathological and molecular analysis, according to the WHO classification [22]. The neurosurgeon collected different samples (tumor or healthy tissue) with image-guidance (Figure 1).

Tissue was defined as healthy if outside the pathological margins, as depicted by preoperative MR available during surgery on a neuronavigation platform (Curve, Brainlab, Munich, Germany). In particular, to collect healthy tissue, the surgeon targeted subcortical sites along the approach for deep-seated lesions, before encountering pathological areas, or at the boundaries of supratotal resection, as depicted by the neuronavigation. Conversely, tumor tissue was collected inside the lesion’s pathological margins, avoiding, when present, necrotic areas, as defined by the neuronavigation.

A sample section deemed surplus to diagnostic requirements was provided immediately after surgical resection to Raman analyst, washed with isotonic NaCl solution and placed under a CaF_2_ window for the analysis. At the end of Raman spectral extraction, samples were sent to a board-certified pathologist for histological and molecular evaluation. The board-certified pathologist was blinded to the RS findings and assessed the nature of the provided sample as either healthy or neoplastic. Each sample was marked as ‘‘Tumor’’ if diagnostic for the presence of glioma cells or ‘‘Healthy” if containing no tumor cells.

### 2.3. Raman Spectroscopic Acquisition

A benchtop RA800 Renishaw spectrometer system (Wotton-under-edge, Gloucestershire, UK) collected the Raman spectra. The system runs a Renishaw’s WiRE 4.0 software and is equipped with a near-infrared (NIR) laser (785 nm) with a maximum power of 500 mW. The initial spectral region of interest was 90–1.800 cm^−1^. We used an exposure time ranging from 0.5 to 2.5 s per spectrum. A range from 1 to 4 spectral accumulations for each acquisition was obtained. The power of the laser was kept at 100% for all the measurements. Spectra were obtained from randomly located points across the sample. The line-focus laser spreads the laser intensity across a line to minimize potential photodamage or fluorescence induction. An automatic performance quality check was performed on silicon and polystyrene internal standards for each measurement to reduce sample-to-sample variation. During the spectral acquisitions, we used the most optimal temperature and humidity conditions and the experimental parameters, such as exposure time, laser power and numbers of accumulations were iteratively modified for each point or area of interest, to optimize the signal and the spectra quality.

For each sample, we sought the maximum number of spectra with a higher signal-to-noise ratio with a punctual acquisition mode. We accomplished the analysis of each sample within 60 min from resection to minimize biological and chemical changing and to best simulate an in-vivo analysis [23].

### 2.4. Data Processing

Pre-preprocessing to remove signal’s baseline drifts [24], cosmic rays and background artefacts [25] was performed before model classification. The data preprocessing (Matlab 2019b Mathworks, USA) followed these steps: (1) truncation of the spectral range to 400–1700 cm^−1^; (2) removal of saturated signal from the dataset using median filtering; (3) selection of spectra with Signal-to-Noise (S2N) ratio ≥3.5 around the phenylalanine 1004 cm^−1^ peak (4) background signal subtraction via Vancouver Raman Algorithm (VRA) method [26]; (5) outlier removal using interquartile range method and (6) signals normalization via root-mean-square (RMS). The resulting number of spectra for the classification were respectively 752 for healthy samples and 665 for tumors samples. (Figure 2) reported the normalized mean spectra with standard deviation for each group.

The wavenumbers of all visible Raman peaks and slopes were identified on the mean Raman plots and compared to those previously reported in the literature, if present, to enhance the reliability of the current findings. We also recorded new prominent peaks at wavenumbers that were not present in the previous study on healthy vs. tumor tissue. We identified 135 Raman frequencies, and we extracted the intensity for each group as input features for the classification algorithms. The algorithms further refined the selection by identifying the minimum number of RS wavenumbers allowing to discriminate between the two groups with satisfying performance thresholds.

Two algorithms developed using Scikit-learn (http://scikit-learn.org, accessed on 26 October 2020) were used to assess classification performance with Leave-one-patient-out cross-validation: Random Forest (RF) and Gradient boosting trees (GB). A threshold between classification performance and robustness was achieved via hyperparameter tuning for each algorithm, along with a statistical feature selection (SciKitLearn FClassif—ANOVA test—top 60 features). An initial hyper-parameters optimization was performed for each classifier in a LOPO cross-validation. Each classifier was validated using a grid-search and 5-fold cross-validation. Parameters for the RF were set: (6,8,10,12) for the maximum tree depth, (50,100,150,200,250,300,350,400,450) for the maximum number of trees. Grid-search parameters for GB were set: estimators (100,150,200), max depth (5,8,10). As results of the optimization, optimal global parameters for classifiers were: RF numbers of trees: 150, max depth 5; while for GB were: numbers of estimators: 200 and max depth 5, learning rate at 0.1.

### 2.5. Statistical Analysis

Statistical analysis was performed among a subset of the top 60 Raman shifts provided by the classification algorithm to define potential new Raman shift useful to evaluate glioma tumoral tissue vs. healthy tissue. A Mann-Whitney test (two-tailed, α = 0.05) was performed, after checking normality using the Shapiro–Wilk test. (software SPSS statistics 24.0; IBM SPSS Inc., Chicago, IL, USA).

## 3. Results

We collected 3450 spectra from 63 different specimens: 1377 labelled as healthy and 2073 labelled as tumor (Table 1).

The results are reported in three sections. The mean spectra of the two groups were initially screened (i) to detect significant Raman peaks with possible biological relevance. The two classification algorithms (GB and RF) employed the intensity of the peaks identified at each shift as discrimination features. These algorithms were then (ii) tested for their discriminative power between neoplastic and normal tissue according to these features. Diagnostic accuracy, sensitivity, and specificity were computed. Finally, the algorithms identified 60 Raman peaks with the best discrimination power between the two groups. These 60 spectra were screened (iii) to analyze potential new peaks as compared to the published literature [5,19,27,28,29,30,31,32,33,34,35].

### 3.1. Classification Features and Algorithms

The algorithm analyzed the 135 Raman Peaks identified at the step (i), and the cross-validation loop identified 60 Peaks with the highest statistical significance and able to provide the highest performance in the two groups distinction. (Table 2) reported Raman Peaks with their biological assignments.

The GB method displayed better performance in distinguishing neoplastic from the normal tissue, with an accuracy of 83%, precision of 82%, recall 82% and F1_score 82%, computed based on tumor detection outcomes of True Positive (TP), True Negative (TN), False Positive (FP), and False Negative (FN). Conversely, The RF method displayed accuracy of 80%, precision of 79%, recall 80% and F1_score 80%.

Receiver Operating Characteristic (ROC) graphs were used to select models for their performance with respect to the False Positive Rate (FPR) and True positive Rate (TPR), and to compute the Area Under the Curve (AUC) to characterize the performance of the classification model (Figure 3). The ROC graphs showed that GB was the best to discriminate normal from tumor samples (all grades and types of glioma), with AUC of 0.82 respect to 0.80 of RF.

Table 3 showed a summary of the performance of the classification models.

### 3.2. Spectral Analysis

The spectral analysis from the Raman active functional groups of nucleic acids, proteins and lipids allowed a thorough characterization of the neoplastic and normal brain biopsies. The average Raman spectra showed differences in the molecular signature between neoplastic and normal specimens (Figure 2). These differences are consistent with previous results from ex-vivo tissue samples [19,27,28,31,32,33,34,35,36].

The band 1003 cm^−1^ reflects the phenylalanine content of the sample as well as the band at 1030 and 1031 cm^−1^. In both groups, we easily recognize very intense bands around 1300, 1440 and a less intense peak around 1660 cm^−1^. These bands are related to protein and lipid. Other bands are visible between 1000 and 1100 cm^−1^, such as band at 1064 cm^−1^ (phospholipids and cholesterol stronger in normal than tumoral tissue), at 1080–1090 cm^−1^ (nucleic acids) and at 1126–1133 cm^−1^ (phospholipids (side-chains) and cholesterol). At lower wavenumbers the cluster of peaks between 419 cm^−1^ and 740 cm^−1^ showed contribution from cholesterol, cholesterol ester (427–430 cm^−1^, 457 cm^−1^, 545 cm^−1^) and choline (719 cm^−1^). Tryptophan contribution at 563 cm^−1^ is usually higher in glioblastoma and oligodendroglioma [37]. The 720:701 cm^−1^ intensity ratio of the mean spectra is increased in Tumor group (0.93 vs. 0.88) and probably reflecting the contribution of glioblastoma biopsies of the samples [32]. Raman bands related to heme are visible at 743 cm^−1^ while in 830, 850 cm^−1^ and 880 cm^−1^ are visible peaks respectively related to DNA, glycogen and tyrosine that are usually higher in high-grade tumors than in normal tissue.

The regions around 1300 cm^−1^ (*p* < 0.001), 1439–1441, and 1740 cm^−1^ (*p* < 0.001) are reduced in Tumor group and are related to fatty acids, CH2/CH3 deformation of lipids side chains, proteins, amino acids, cholesterol/cholesterol ester [30] usually present in normal with matter and ester group (1740 cm^−1^). The band at 1337 cm^−1^ is less intense in the Healthy specimens (*p* < 0.001) and is related to aliphatic amino acids, including tryptophan, nucleic acids and glycogen found in necrosis. The bands at 1003 cm^−1^ (*p* < 0.001) and 1031 cm^−1^ (*p* < 0.001) significantly different in phenylalanine in the two groups that are usually higher in glioma than normal tissue and astrocyte [30,38]. An increased DNA content related to band intensities at 498 cm^−1^ (*p* < 0.001), 830 cm^−1^ (*p* < 0.001) and 1087 cm^−1^ (*p* < 0.001) was also found in the Tumor group. A reduced lipid/cholesterol content on bands at 419, 427–430, 615, 968 cm^−1^ (all *p* < 0.001) is seen in the Healthy group. Intensities of bands related to proteins were significantly different (826, 880, 881, 883, 927, 934, 950, 963, 1035, 1583 1603–1616 cm^−1^ and 1660 cm^−1^, *p* < 0.001) between the two groups underling a changed protein profile related to tumoral proliferation.

### 3.3. Novel Raman Shifts

Among the top 60 Raman shifts features provided by the classification algorithm of the current study as highly significant in the distinction between normal and neoplastic tissues, 28 were previously described [5,8,11,17,18,19,33,39,40,41,42,43,44,45,46,47], corresponding at the following shifts: 457, 540, 754, 826, 850, 853, 880, 881, 883, 903, 927, 928, 934, 950, 958, 977, 1003, 1030, 1122, 1337, 1578, 1581, 1604, 1614, 1616, 1657, 1659, 1660 cm^−1^ (Table 2). We thus focused on the new 19 Raman Shifts with documented biological significance. These 19 bands were not used before in discriminating between glioma and healthy tissue. The intensity of Raman shift at each band resulted different between the two classes of samples, i.e., between tumor and normal tissues, underlining their biological difference [29,30]. The analysis confirmed statistically significant differences (*p* < 0.001) between normal and neoplastic tissue biopsies in the new 19 Raman shifts related to proteins (524, 933, 963, 1031, 1035, 1583, 1603 cm^−1^), nucleic acids (498, 780, 825, and 894 cm^−1^), lipids (431, 776, 875, 968 cm^−1^), collagen (817 cm^−1^), glycogen (941 cm^−1^), heme content (743 cm^−1^) and calcification (975 cm^−1^) (Figure 4).

## 4. Discussion

This study demonstrated that a benchtop Raman Spectroscopy machine could distinguish between normal and neoplastic tissue in fresh tissue samples ex-vivo with good accuracy, sensitivity, and specificity when paired with supervised machine learning techniques. These findings added new evidence to preliminary results provided by the few available studies [5,18,42]. The study also identified 19 bands that had not been previously described that were different between pathological and normal tissue. About the two machine learning classification models, the GB had a better performance in classifying the tissue samples.

The balance between the patient’s functional integrity and the extent of tumor resection is fundamental: non-invasive tools providing rapid tissue assessment are thus advocated. Surgery is one of the most important prognostic factors in treatment as evidences that a more significant surgical safe resection leads to improved outcome are continuously emerging [20]. These goals cannot always be achieved due to the infiltrating behavior of gliomas. The challenge of contemporary neuro-oncological surgery is also to find the proper balance between tumor resection and the patient’s functional integrity [20] and to detect infiltrating neoplastic cells beyond pathological margins; an intra-operative tumor detecting technology can thus be strategic to enhance surgery, especially at the limit of resection where tumor infiltration can still be present but a level lower than the tumor core and the detection threshold of available methods. The decision regarding a further resection is to be critically balanced with an increased risk of new-onset neurological morbidity, that is more likely just at this limit, where surgery is at closer proximity to eloquent structures.

### 4.1. Raman Spectroscopy with Fresh Tissue Samples in an Ex-Vivo Scenario

The Raman system was next to the operating room, and the spectral measurements were acquired within 60 min from multiple regions of the fresh tissue immediately after surgical resection. The acquisition time was typically on the order of seconds for a single spectrum. No samples processing or manipulation was performed before optical analysis. In-vivo tissue discrimination between healthy and tumoral tissue needs data from fresh samples that can make Raman analysis more precise and capable to characterize the tissue even in the presence of biological contaminants and with a low number of total spectra acquired, to keep analytical time at minimum and be compatible with a regular surgical workflow. During surgery time is limited for spectral acquisition: an analytical algorithm able to extract as much information as possible from Raman spectra with reliable accuracy can thus improve the detection process. This study is among the few [5,18,42] available where untreated and unfrozen tissue is analyzed to best approximate an intra-operative scenario. The data from cryosection and FFPP tissue blocks can contain different biological artefact [19,48]. The presence of freeze artefact in samples can low the discrimination power, hampering an accurate analysis and changing the molecular structure of the biopsies [19]. Yet, frozen sections are currently more available and relatively rapid, but they can also be hampered by the compulsory availability of neuropathologic expertise. If robust, the RS herein reported has the potential to be automated and deployed directly by brain surgeons.

This study, consistently with previous results [49], demonstrated that Raman spectroscopy could detect changes in the chemical composition and molecular structure between pathological and healthy fresh tissue. The spectral characteristics of tumoral and healthy tissue properly matched with those established on the topic [49]. Machine learning algorithms allowed to find 60 best representative Raman shifts among over 135 bands analyzed.

### 4.2. Characterization of Raman Shifts

Out of 60 bands, 28 had already been previously described as different between healthy and tumor samples [5,8,11,17,18,19,33,39,40,41,42,43,44,45,46,47]. In particular, distribution in relative intensity of these bands was related to well-known differences in the biology and structural composition of tumoral and brain tissue such as DNA Amide I, protein and calcification that can be seen in the necrotic tumor core. Among these 60 top characterizing Raman bands, new shifts related to proteins (524, 933, 963, 1031, 1035, 1583, 1603 cm^−1^), nucleic acids (498, 780, 825 cm^−1^, and 894 cm^−1^), lipids (431, 776, 875, 968 cm^−1^), collagen (817 cm^−1^) and calcification within the samples (975 cm^−1^) were identified and characterized as different between healthy and glioma biopsies. Only once peaks at 498, 875, 934 cm^−1^ and 1031 cm^−1^ were previously reported [31], yet from the analysis of 6 frozen glioma samples and with no comparison with healthy tissue. The biological importance of these bands in gliomas samples is therefore strengthened by the findings of the current study. In the future, authors could include these 19 Raman shifts that are all related to well-known biological compounds with coherent biological differences between healthy and neoplastic tissue to improve the accuracy of Raman classification methods in untreated tissue and in an in-vivo scenario.

### 4.3. Diagnostic Performance of Machine Learning Classification Models

The RF and GB displayed an overall good performance and allowed to distinguish tumor from the normal brain with an accuracy respectively of 80% and 83%. This performance matched that reported in previous works [13,41], ranging from 85% to about 99%. However, it resulted lower to studies done in a similar ex- and in-vivo setting with untreated tissues to discriminate between tumor and normal tissue [5,11,18]. The performance is also lower than that obtained from datasets on frozen and FFFP section, where, however, a longer preparation step can make a potential intra-operative application less likely and alter the biochemical properties of the sample. Analyzing the distinct pre-analytical and analytical phases of our workflow, we collected evidence leading to arguing that well-controlled acquisition setting ex- and in-vivo and the tissue conservative status are an essential preprocessing step to grant optimal signal acquisition and, thus, final analytical output. The differences identified by the employed algorithms between the normal and the neoplastic tissues resulted consistent with previous findings, as for what machine learning as a separate step is concerned. The latter argument contributes to increasing the likelihood that this technique and workflow can be effectively reproduced.

### 4.4. Study Limitations

The time requested for analysis is higher than that previously described [5]. However, a learning effect was observed during the study, with lower analytical time with later samples. Data regarding diagnostic power in discriminating relevant molecular features are lacking [50] and should also be addressed to probe the diagnostic accuracy in this relevant issue. The limited number of cases also hamper a broader analysis of the data. However, given the few clinical series currently reported, even such a small clinical cohort can still represent a useful contribution to the field. It should also be remarked that no specific marker exists in IDH-wild type gliomas that absolutely rule out a single-cell infiltration; the definition of normal tissue should be deployed accordingly. Although developed in a real surgical scenario, this study was not powered and designed to assess how RS can impact the surgical practice itself and whether the RS can actually modify the surgical behavior, such as yielding to broader resection, a better outcome with lower morbidity and improved detection of neoplastic infiltration, for instance during recurrent disease after previous multimodal treatment or with non-enhancing gliomas, that show no intra-operative fluorescence [51].

We aimed at providing preliminary technical and clinical advances on dealing with untreated tissue, to explore whether this translational study can lead to further development of an intra-operative technology that could be considered safe, accurate and reliable to be ultimately deployed in-vivo. Future in-vivo studies should address these issues, evaluating the impact on surgical resection and outcome, upon findings provided by this and similar studies [5].

## 5. Conclusions

RS reliably discriminated glioma tumor and normal tissue on fresh tissue. Nineteen new Raman peaks useful to discriminate between glioma and healthy tissue were identified. The analysis of these new bands can support further development of real-time tissue analysis, increasing the accuracy and efficiency in a neurosurgical scenario. This study added a robust contribution to the application of this technology to oncological brain surgery. A refined investigation of the biochemical correspondences between the Raman results and the molecular biology of the lesion could provide insights into gliomagenesis and infiltrative behavior.

## Figures and Tables

**Figure 1 cancers-13-01073-f001:**
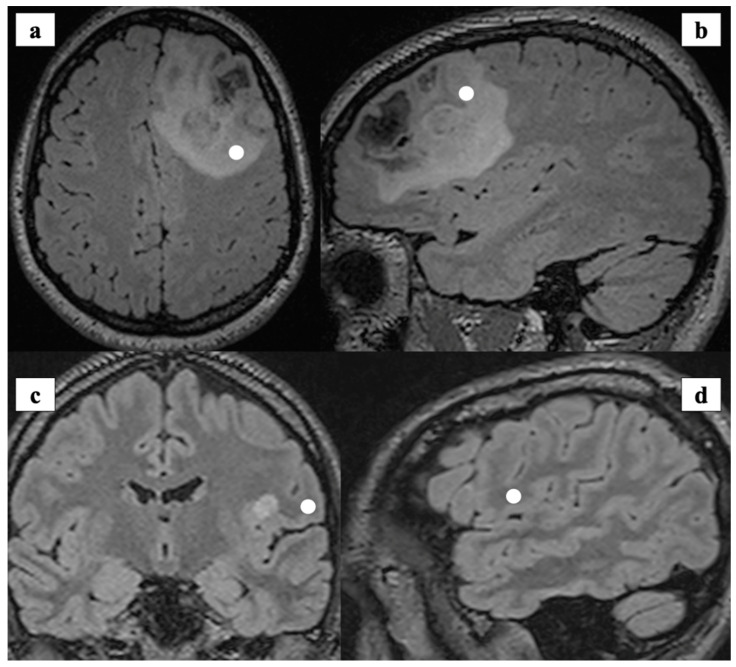
(**a**,**b**): Preoperative MRI (axial and sagittal view) showing tumor consistent with Anaplastic Oligodendroglioma IDH-1 mutant, 1p/19q co-deleted grade III (WHO 2016). The white spots show the intraoperative site of tissue biopsy registered with Neuronavigation system and labelled as *tumor*. (**c**,**d**): Preoperative MRI (axial and sagittal view) showing the intraoperative location of tissue collection that was labelled as *healthy*. The specimen was collected (white spot) and analyzed during the surgical approach to a peri-insular ganglioglioma (Grade I, WHO 2016).

**Figure 2 cancers-13-01073-f002:**
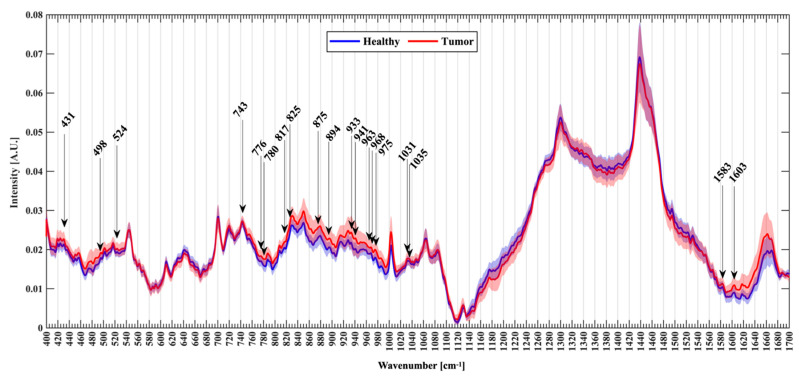
Normalized mean spectra with standard deviation for healthy (blu) and tumor patients (red). Arrows mark the new Raman peaks identified.

**Figure 3 cancers-13-01073-f003:**
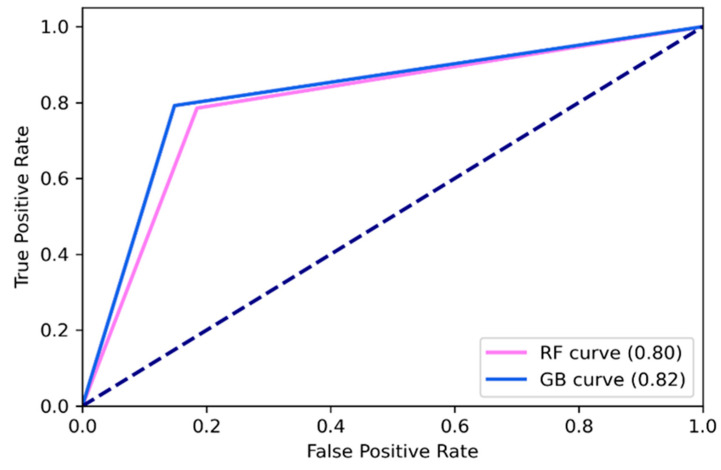
Receiver operating characteristic curve for Random Forest (RF) and Gradient boosting trees (GB).

**Figure 4 cancers-13-01073-f004:**
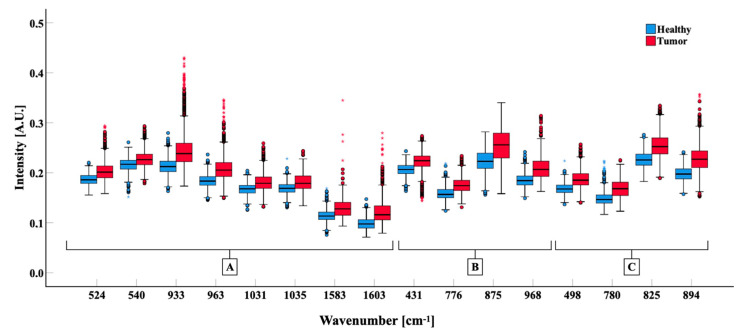
Comparison of the band intensities of Healthy (*blue*) and Tumor (*red*) biopsies grouped for biological significance. Box-plots show 25°, 50°, 75° percentiles (box) and 5°–95° (whiskers) percentile of data distribution; the median of groups is indicated with a black bar. Mann–Whitney test was calculated for all bands showed (*p* < 0.001). (**A**) bands related to protein. (**B**) bands related to lipids. (**C**) Bands related to nucleic acids.

**Table 1 cancers-13-01073-t001:** Numbers of specimens and histological diagnosis. *n*°: numbers.

	***n*° of Samples**	***n*° of Spectra**
**Glioma tissue**	38	2073
**Grade II**		
Astrocytoma, WHO grade II (IDH-mutant)	0 (0)	-
Oligodendroglioma, WHO grade II (IDH-mutant)	4 (10.5)	220
**Grade III**		
Astrocytoma, WHO grade III (IDH-mutant)	5 (13.2)	259
Oligodendroglioma, WHO grade III (IDH-mutant)	12 (31.6)	642
**Grade IV**		
Glioblastoma, WHO grade IV (IDH wild-type)	14 (36.8)	773
Glioblastoma, WHO grade IV (IDH mutant)	3 (7.9)	179
**Normal Tissue**	25	1377
**Total**	63	3450

**Table 2 cancers-13-01073-t002:** Tentative assignment of the main Raman shifts according to previous studies [5,18,26,27,28,29,30,31,32,33,34,35,36,37].

Raman Shift (s^−1^)	Proposed Assignments
431 *	Cholesterol/cholesterol ester
450	Ring torsion of phenyl
457	Proteins and cholesterol
460	Undefined
474	Glycogen and polysaccharides
478	Polysaccharides
495	Undefined
498 *	Nucleic acids/nucleotides
502	Undefined
517	Undefined
524 *	S-S disulfide stretching in proteins
540	n(S-S) trans-gauche-trans(amino acid cysteine)
743 *	Heme (blood)
754	Symmetric breathing of tryptophan
776 *	Phosphatidylinositol
780 *	Uracil based ring breathing mode
808	Undefined
817 *	C-C stretching/collagen assignment
825 *	Phosphodiester
826	Tyr, proline
837	Undefined
850	Glycogen (high-grade tumors)
853	Glycogen (high-grade tumors)
875 *	Choline and phospholipids
880	Tryptophan, d(ring)
881	Hydroxyproline and tryptophan (collagen); sterol ring stretch of cholesterol; asymmetric stretching of choline
883	ρ (CH2) (protein assignment)
893/4 *	Phosphodiester (nucleic acids)
903	Undefined
916	Undefined
927	Undefined
928	Amino acids proline & valine (protein band)
933 *	Proline, hydroxyproline
934	C-C backbone (collagen assignment)
941 *	Glycogen
950	single bond stretching vibrations for the amino acids proline and valine and polysaccharides
954	Undefined
958	Stretching vibrations of PO_4_ in hydroxyapatite
963 *	Protein assignments
968 *	Lipids
975 * and 977	Tricalcium phosphate Ca3(PO4) calcification seen in schwannoma and necrosis
1003	Ring breathing mode of phenylalanine of protein
1030	Phenylalanine (protein assignment) and collagen
1031 *	Phenylalanine (protein assignment)
1035 *	Collagen
1036	Undefined
1122	Glycogen
1337	Aliphatic amino acids, including tryptophan, nucleic acids. Glycogen in necrosis
1342	Aliphatic amino acids, including tryptophan, nucleic acids. Glycogen
1578	C-C stretch of protein, nucleic acids
1581	C-C stretch of protein, nucleic acids
1583 *	C-C stretch of protein, phenylalanine, nucleic acids
1603 *	Phenylalanine & tyrosine/Oxygenated haemoglobin
1603	Cytosine, tyrosine, phenylalanine
1614	Aromatic amino acids (protein); Tyrosine e proline
1616	C-C stretching mode of tyrosine & tryptophan
1657	Amide I (a-helix) (protein)
1659	Amide I (a-helix)
1660	Amide I band

Asterisk (*) marks new Raman peaks used for the first time for discrimination between glioma and healthy tissue in fresh samples.

**Table 3 cancers-13-01073-t003:** Results of classification models for normal vs. glioma tissue biopsies using Raman Spectroscopy.

Performance Metrics	Random Forest	Gradient Boosting
Accuracy	0.80	0.83
Precision	0.79	0.82
Recall	0.80	0.82
F1-score	0.80	0.82

## Data Availability

The datasets generated during and analyzed during the current study are available from the corresponding author on reasonable request.

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
