# Peer review of "Glioma *biopsies* Classification Using Raman Spectroscopy and Machine Learning Models on Fresh Tissue Samples"

_cancers, 2021, doi:10.3390/cancers13051073_

Round 1
Reviewer 1 Report
The authors answered all the questions of the reviewers, made significant changes to the article. In its current form, the article can be recommended for publication.
Author Response
We thank the reviewer for appreciating our study.
Reviewer 2 Report
The exact writing should be:
Astrocytoma, WHO grade II
Author Response
We thank the reviewer for appreciating our study.
We edited the writing about Astrocytoma as suggested.
Reviewer 3 Report
It appears that they addressed my questions and comments in the cover letter and manuscript. I am fine with their edits and would vote to publish.
Author Response

(The authors gave the same response as above.)

Reviewer 4 Report
The authors addressed almost all of my issues raised and improved the manuscript significantly. Only one point - the most important one - remains valid:
“Authors pointed out, that their approach led to significantly lower accuracies than reported in different studies (discussion 4.3; 83% vs up to 99%).Either authors should test their approach on the data obtained by other research groups or using previously published classification approaches on their data to demonstrate comparable results. This validation step is of very high importance, because otherwise the only gain of the manuscript would be the newly identified peaks that might help to improve glioma detection, but could not be used in such a way to actually improve accuracy in discrimination between healthy and glioma tissue. Without this additional validation the overall gain of knowledge of the given manuscript would be too low to justify publication in “cancers”.”
The authors explained conclusively that there is no additional test-data available from other studies to test their algorithms, but the main issue that classification accuracy is significantly below a previously reported accuracy for a study that also uses fresh brain and brain-tumor samples remains unaddressed. The mentioned study also used boosted decision trees (Jermyn et al 2015, doi: 10.1126/scitranslmed.aaa2384), the same approach used here, but is getting significantly higher accuracies (92%) with a lower number of patient samples. While I do acknowledge the significantly increased sample size, the authors did not provide sufficient evidence that the additional Raman peaks identified actually help differentiating between tumor and healthy tissue. Consequently, the question remains what unique new insight this manuscript does provide.
I highly recommend to the authors either to improve their classification approach, using additional features, etc. or using only the features used by Jermyn et al (Jermyn et al 2015, doi: 10.1126/scitranslmed.aaa2384) on the here obtained data to demonstrate that the identified new Raman peaks are providing additional information for tumor classification. One way or the other authors have to give a convincing explanation for the significant differences in classification goodness.
Additional minor issues:
Figure legend of figure 4: I suppose p<0.001 is meant? First line: an “and” is missing. Explanation of “A)” contains an error.
Round 2
Reviewer 4 Report
The authors adressed all issues raised. I would like to congratulate the authors to the very good study!
This manuscript is a resubmission of an earlier submission. The following is a list of the peer review reports and author responses from that submission.
Round 1
Reviewer 1 Report
The article is devoted to an actual problem, well written, logical and structured. As for the study design, I have no questions or comments. It is interesting why the authors chose and limited themselves to only Raman spectroscopy. In the study of gliomas, good results were obtained using FTIR spectroscopy. In particular, in the same journal in 2020 a work was published (Cancers 2020, 12, 3682; doi: 10.3390 / cancers12123682), in which 79 patients with gliomas were divided into the positive class (IDH1-mutated) from the IDH1- wildtype glioma, with a sensitivity and specificity of 82.4% and 83.4%, respectively. The reviewed work does not mention such possibilities. Is it possible to combine both methods? If so, what prospects does this open up? Please discuss these points.
Reviewer 2 Report
The study „Glioma biopsies classification using Raman spectroscopy and machine learning models on fresh tissue samples” by Riva et al uses Raman spectroscopy to differentiate between fresh, primary glioma samples and healthy brain tissue, identifying several new Raman peaks facilitating differentiation between both tissue types. While the study is principally of very high interest to the readers of “cancers” and is scientifically sound in many aspects, the manuscript in the given form lacks several important methodological descriptions and validations, as well as a more detailed comparison to the literature. Most importantly, a validation of the classification model or the obtained data is necessary, as authors noted themselves that the reported predictive power of the used models is significantly lower than previously reported ones. This validation step is mandatory; as otherwise, the manuscript does not provide sufficiently new insights to justify publication in cancers. Furthermore, the description of the data and the figures need to be reworked. Thus, I recommend a major revision. Points of criticism are described below in more detail.
Major concerns:
1) As authors also noted the point of tissue sampling is important, as this may alter the results. Please specify the location of the biopsies in more detail.
2) Line 123-125: Authors should describe parameter settings in more detail, so others can reproduce the results.
3) Line 142: Please describe in more detail how the Raman peaks were selected. Which criteria/algorithms were used?
4) Authors used random forest and gradient boosted decision trees for classification. Would different machine learning approaches, such as SVM, naïve bayes, discriminant analysis, etc. yield better results? Were they tested? If so, please report it within the results. If not, please explain the reason for the choice of model used in the manuscripts, as decision trees are sub-par in terms of generalization. In addition, the precise hyperparameters of the used models need to be added to allow proper reproduction and comparison for future studies.
5) Figure 4: Needs a full rework, as e.g. axis labels of B and C are barely visible and the style/size of the sub-figures is not uniform.
6) Authors pointed out, that their approach led to significantly lower accuracies than reported in different studies (discussion 4.3; 83% vs up to 99%). Either authors should test their approach on the data obtained by other research groups or using previously published classification approaches on their data to demonstrate comparable results. This validation step is of very high importance, because otherwise the only gain of the manuscript would be the newly identified peaks that might help to improve glioma detection, but could not be used in such a way to actually improve accuracy in discrimination between healthy and glioma tissue. Without this additional validation the overall gain of knowledge of the given manuscript would be too low to justify publication in “cancers”.
Minor concerns:
7) Simple summary: introduce abbreviations before first usage (line 23).
8) Check for spelling mistakes and other errors, e.g. the “7” in line 68 or “with” in line 99 should be white, “cm-1” in the text should be cm-1, sometimes no units are used, etc
9) How long does the prediction step for a new tumor sample take?
10) Table 1: Please list the number of spectra obtained for each tumor entity.
11) Table 1, line 162: The meaning of the line is unclear.
12) Figure 3: Please include color-coding in the figure itself. The x-axis shows the wrong unit. Please mark the bands mentioned in the results-section in the graph. Otherwise, it is very laborious to follow the text.
Reviewer 3 Report
Thank you for this submission; here, you describe the generation of a Raman spectroscopic profile from fresh samples of healthy and cancerous brain tissue and the application of a machine learning protocol to those profiles. Clearly, the intraoperative assessment of brain tumors presents opportunities for improvement. However, in my opinion some points in the data presentation and discussion merit some further exploration.
First, there is some mention as to the significance of some of the spectroscopic shifts between healthy and diseased tissue. However, diseased tissue can comprise widely different components. Some parts of (particularly high grade) gliomas are largely necrotic, and other areas can have prominent vascularity. All infiltrating gliomas can show marked variation in terms of cellularity. I would like to know a bit more detail on the makeup of the different tissue samples, though I know the study design did not readily explore this detail upfront. It could be useful to delve deeper though, even if only to demonstrate that you looked into it and performed some comparison to histopathological parameters. Did you consider applying Raman technology to generate recapitulated images as well as spectra? Such a comparison may have been fruitful.
You do cover that this investigation was not designed to assess impact on current practice. It may be helpful to be clearer that currently, intraoperative frozen sections are relatively rapid (20 minute turnaround time in many centers) and that frozen tissue is not necessarily damaged by exposure to solvents or formalin at the time of frozen section. They are hampered, though, by the availability of neuropathologic expertise and the mentioned tissue artifacts. The proposed method would be easier to automate and usable directly by neurosurgeons.
One other item - a ganglioglioma is shown in Figure 1 but not mentioned elsewhere to my knowledge. Was this an oversight?
Reviewer 4 Report
The authors determined by Raman spectroscopy differences between brain tumors and "normal" brain tissue adjacent to the tumor on fresh tissue samples.
Several points should be addressed:
- The authors should not use the term Cancer but rather Brain tumor
- Grade in Tabel 1 reads WHO grade
- spectra differed between tumor and non-tumor; however the biologic substrate hiding behind a spectrum is not well known. Could you clarify soem aspects in oligodendrogliomas where you report Raman shifts with regard to calcifications.
- Choline can be measured by MR spectrocopy. Did you try to correlate thus data with RS?
- The english style need serious improvement. One does not understand what is meant in the sentence 342-345. Further exapmles are found in line 333 (it resulted slightly), 336 (obtain), 268 (different), 278 (patience), 204 (throughout), 68 (7)
